# Protective Effects of Huo Xiang Zheng Qi Liquid on *Clostridioides difficile* Infection on C57BL/6 Mice

**DOI:** 10.3390/microorganisms12081602

**Published:** 2024-08-06

**Authors:** Ming Chen, Lin Zhai, Kristian Schønning, Warner Alpízar-Alpízar, Ole Larum, Leif Percival Andersen, Susanne Holck, Alice Friis-Møller

**Affiliations:** 1Department of Clinical Microbiology, University Hospital of Southern Denmark, 6200 Aabenraa, Denmark; 2Department of Clinical Microbiology, Copenhagen University Hospital, Rigshospitalet, 2100 Copenhagen, Denmark; 3Department of Clinical Microbiology, University Hospital of Copenhagen, Hvidovre, 2650 Hvidovre, Denmark; 4The Finsen Laboratory, The Finsen Center, Copenhagen University Hospital, 2200 Copenhagen, Denmark; 5Department of Pathology, University Hospital of Copenhagen, Hvidovre, 2650 Hvidovre, Denmark

**Keywords:** *Clostridioides difficile*, *C. difficile*-associated disease, vancomycin, Huo Xiang Zheng Qi

## Abstract

Background: *Clostridioides difficile*-associated disease (CDAD) is a major healthcare-associated infection. New treatment options for CDAD are needed. A traditional Chinese medicinal formula, Huo Xiang Zheng Qi (HXZQ), was chosen to test against CDAD in a mouse model. Methods: C57BL/6 mice were challenged with *C difficile* (ATCC 43255) orally; then received saline; vancomycin 25 mg/kg; or HXZQ in two different concentrations twice daily for 5 days. The animals’ body weight; clinical signs; and survival rates were registered daily. Fecal pellets from each animal were taken for PCR analysis as a control of infection. Results: 50% of the mice receiving saline died; 85.7% of the mice receiving vancomycin survived; 75% of the mice receiving HXZQ survived; and 87.5% of the mice receiving a 1:1 saline dilution of HXZQ survived. The HXZQ-treated groups were *C. difficile* PCR positive with loads less than that of the untreated mice. The weight loss in the vancomycin plus HXZQ 1:1 treated group; the vancomycin-treated group; and the untreated group were 3.08%, 4.06%, and 9.62%, respectively. Conclusions: our results showed that HXZQ can protect mice from CDAD-related death as effectively as vancomycin and the combination of vancomycin and HXZQ may give even better protection.

## 1. Introduction

*Clostridioides difficile* is an anaerobic Gram-positive rod carried in the guts of 2–7% of healthy human adults harmlessly [1]. However, when an individual takes an antibiotic for another condition, *C. difficile* can grow and replace most of the normal bacterial flora living in the colon [2]. The excessive proliferation of *C. difficile* results in an infection (CDI) characterized by symptoms such as diarrhea, fever, and pain. CDI is the predominant infectious cause of healthcare-acquired diarrhea globally and can lead to extended hospitalization [1,3,4]. It primarily affects elderly patients with comorbidity in whom the intestinal flora has been disrupted by the previous use of antibiotics [5,6,7].

A number of herbal medicines have been used to treat diarrhea in Traditional Chinese Medicine. A classical formulation, consisting of a mixture of various plant extracts known as “Huo Xiang Zheng Qi Liquid (HXZQ)”, has been extensively used to treat a range of common gastrointestinal infectious diseases, including acute gastroenteritis and functional dyspepsia, since the Song Dynasty. It is documented in the *Prescriptions People’s Welfare Pharmacy* (Taiping Huimin Hejiju Fang) published by the Imperial Medical Department of the Song Dynasty in 1078. HXZQ is composed of 11 herbs: *Arecae pericarpium* (Dafupi), *Angelicae Dahuricae Radix* (Baizhi), *Perillae Folium* (Zisuye), *Poria* (Fuling), *Pinelliae Rhizoma* (Banxia), *Rhizoma Atractylodis Macrocephalae* (Baizhu), *Citri Reticulatae Pericarpium* (Chenpi), *Magnoliae Officinalis Cortex* (Houpu), *Radix Platycodonis* (Jigeng), *Herba Pogostemonis* (Huoxiang), and *Radix Glycyrrhizae* (Gancao). It has been reported that HXZQ can promote gastroenteric motility, ameliorate myoelectricity of enteric smooth muscles, enhance intestinal mucosal protection, and regulate gastrointestinal function [8,9,10]. He et al. have shown that HXZQ can reduce diarrhea in BALB/c mice with *Bacillus dysenteriae* and *Salmonella typhimurium*-induced diarrhea with the regulation of CD4^+^ and CD8^+^ cells in Peyer’s path and suppression of TNF-α levels in the enteric mucosa [11]. Li et al. showed that HXZQ can prevent acute intestinal injury induced by heatstroke in rats [12]. Liu et al. showed that HXZQ has a protective effect on the intestinal mucosal mechanical barrier in rats with post-infectious irritable bowel syndrome induced by acetic acid [13]. A recent study showed that HXZQ exhibits modulatory effects on gut-microbiome homeostasis in healthy adults and antibiotic-induced gut-microbial dysbiosis mice mode [14]. All these studies indicate that HXQZ may be helpful in treating CDI. The aim of this study was to investigate HXZQ treatment alone and in combination with vancomycin treatment in a mouse CDI model and to compare their efficacy to standard vancomycin treatment.

## 2. Materials and Methods

### 2.1. Bacterial Strain

*C. difficile* strain ATCC 43255 encoding enterotoxin A and B, but not binary toxin (*tcd*A+, *tcd*B+, *cdt*B− was used in both in vitro and in vivo studies. The strain was stored in infusion broth with reinforced 10% glycerol (Sigma-Aldrich, St. Louis, MI, USA) at −80 °C until use and was sub-cultured on 5% blood agar plates in an anaerobic atmosphere comprising 10% H_2_, 10% CO_2_, and 80% N_2_ at 37 °C for 24 h. For the *C. difficile* challenge in mice, a suspension of four days of *C. difficile* culture was prepared, and a bacterial count was performed at the time of challenge to ensure that the correct dose of *C. difficile* cells was administered. 

### 2.2. Tested Agents

The Huo Xiang Zheng Qi Liquid (HXZQ) for the study was purchased from Tongrentang Pharmaceuticals Group (GMP certificated), Beijing, P. R. of China (No: 1146200). HXZQ consists of 11 herb extracts and is a liquid with 45% ethanol. All antimicrobial agents and *C. difficile* Toxin A and B were purchased from Sigma-Aldrich (St. Louis, MI, USA).

### 2.3. In Vitro Antibacterial Study

The in vitro study of HXZQ on *C. difficile* ATCC43255 was first carried out in three different agar plates, including (1) a chocolate agar plate, (2) a Mueller Hinton II Agar (MHA) plate, and (3) a SSI 5% horse blood agar plate (5771422) (SSI Diagnostica, Hillerød, Denmark). A suspension of the bacteria in McFarland 0.5 was used in the *in vitro* study. Twenty microliters of vancomycin (20 μg), 45% ethanol, or HXZQ were added on the tops of the discs. All agents were applied to paper discs (20 μL volume) on the tested plates. Vancomycin and ethanol were used as positive and negative controls, respectively. 

### 2.4. Cytotoxicity of Toxin A, B, and HXZQ on the In Vitro Growth of McCoy Cells

McCoy cells were cultured in a RPMI1640 medium with 10% fetal bovine serum (Sigma-Aldrich, St. Louis, MI, USA) at 37 °C and 5% CO_2_. A cell culture was seeded at 2 × 10^5^ cells in 0.2 mL in a well (96-well plate) and incubated overnight. *C. difficile* bacterial supernatant, Toxin A, B, and HXZQ were added into the cell cultures and further cultured for 20 h. The cultures were fixed with trichloroacetic acid and then stained with sulforhodamine B. The optical density (OD) was read at 490 nm by an ELISA reader [15].

### 2.5. C. difficile in Fecal Samples

Before the in vivo treatment study started, 12 mice were infected with *C. difficile* (ATCC 43255). Their body weights were monitored and fecal samples were collected for the detection of *C. difficile* infection. The fecal samples were thawed, dissolved, and incubated in 0.5 mL of 70% alcohol. After 1 h, the samples were diluted 10-fold and plated on cycloserine–cefoxitin fructose agar (State Serum Institute, Denmark). After incubation in an anaerobic atmosphere for 48 h, the colonies were counted. To confirm *C. difficile* in the mice, the isolates were examined with MALDI-TOF-MS (Bruker, Bremen, Germany) and real-time PCR with Applied Biosystems 7500 Fast Real-Time PCR System [16].

### 2.6. PCR Monitoring C. difficile Toxin in Faecal Samples

Fecal samples were analyzed using a multiplex, internally controlled real-time PCR assay for the detection of toxigenic *C. difficile* [16].

### 2.7. Animals and Housing

The animals were 6-week-old C57BL/6 female mice with body weights of approximately 20 g and were obtained from Taconic, Lille Skensved, Denmark. All animal studies were performed at the Department of Experimental Medicine, Copenhagen University. The animals were housed in a biohazard level II facility and provided with autoclaved food, water, and bedding, with 12 h light–dark cycles. The mice were housed at a maximum of four in an individually ventilated cage (Techniplast, Buguggiate, Varese, Italy). 

### 2.8. Mouse Model of CDI and Effect of HXZQ on CDI in Mice

The model used in this study was based on the model developed by Chen, X. et al. [17]. Figure 1 illustrates the experimental scheme. The course of infection in *C. difficile*-infected mice can be divided into two phases: the treatment phase, with an acute course of disease within the first 3 days after challenge, and the post-treatment phase.

Briefly, C57BL/6 female mice received an antibiotic cocktail in drinking water (kanamycin 0.4 mg/mL, gentamicin 0.035 mg/mL, colistin 850 U/mL, metronidazole 0.215 mg/mL, and vancomycin 0.045 mg/mL) for 3 days and then regular drinking water for 3 days. After a 2-day break, they received clindamycin 10 mg/mL in 0.2 mL (in the abdominal cavity). The mice were inoculated with the *C. difficile* ATCC 43255 (enterotoxin A+/B+) strain (2 × 10^4^ CFU) intragastrically one day later (Day 0). 

Each group consisted of eight mice. The animals received 0.1 mL of saline, vancomycin (0.5 mg), or HXZQ 1:1 diluted with saline (Tong Ren Tang Technologies Co., LTD, Beijing, China) in two different concentrations twice daily for 5 days from Day 0. The animals’ body weight, clinical signs, and survival rates were registered daily for 10 days post-CD inoculation. 

The animals with a weight loss above 20% or judged to be in a moribund state based on a clinical score were euthanized. The animal’s body weight, clinical signs, and survival rate were monitored for 10 days post-inoculation. Then, the animals were sacrificed and their coecums and colons were removed for histological examination. Fecal samples from each animal were collected before the inoculation, 3 days after inoculation, and on the day of death or sacrifice and immediately stored at −20 °C for PCR analysis. A multiplex, internally controlled real-time PCR assay for the detection of toxigenic *C. difficile* was used in the study with the previously described method [16]. 

### 2.9. Study of Treatment with HXZQ and Vancomycin Combination on CDI in Mice

The effects of HXZQ and Vancomycin on *Cl. difficile* (ATCC 43255) infection were also studied on C57BL/6 mice. Each of the four groups had eight mice. The animals received 0.1 mL of saline, vancomycin (0.5 mg), or HXZQ in a 1:1 dilution, or HXZQ and vancomycin twice daily for 5 days from Day 0. The animals’ body weights, clinical signs, and survival rates were registered daily for 21 days post-inoculation. The animals were then sacrificed and their coecums and colons were removed for histological examination. Fecal pellets from each group were taken for PCR analysis and culture 3 days and 1 day before CD inoculation, as well as 3, 7, 14, and 21 days after the inoculation, and on the day of death or sacrifice. 

### 2.10. Histopathological Examination

The mice were euthanized by cervical dislocation. Caecum and colonic tissue were fixed overnight in 4% paraformaldehyde and paraffin embedded. Samples from each mouse were coded and randomized. Then, 3-micron tissue sections were deparaffinized in xylene and hydrated in a gradual series of ethanol–water dilutions. Sections were stained with hematoxylin and eosin (H&E) and examined by light microscopy. The histological severity of colitis was graded using a scoring system reported previously [17,18]. It was used to determine epithelial damage (score 0–4), edema (score 0–4), and cellular infiltration (score 0–4) and assigned a total score between 0 and 12. An experienced histopathologist, blinded to the treatment, evaluated the slides and all the tissue on the slides. 

### 2.11. Ethics Statement

This study was conducted with the approval of the Danish Veterinary and Food Administration (J.nr. 2007/561-1394) and the Danish Working Environment Authority (J.nr. 20110073605/2). All recommended guidelines for the care and use of animals were followed. To ensure that no mice suffered during the experiment, animals with a weight loss above 20% or judged to be in a moribund state based on the clinical score were euthanized. The working area was sterilized using 2% Virkon S between treatment groups to prevent cross contamination.

## 3. Results

### 3.1. In Vitro Study 

HXZQ showed no antibacterial activity, as it showed the same inhibition as 45% ethanol on the in vitro growth of *C. difficile* ATCC43255 (Table 1). *C. difficile* ATCC43255 bacterial supernatant and HXZQ 1:64 inhibited the in vitro growth of McCoy cells by 31 and 24%, respectively (Table 2). *C. difficile* Toxin A and B inhibited the in vitro *growth* of McCoy cells, by 46 and 39%, respectively. HXZQ (1:128 dilution) did not reduce the cytotoxicity of Toxin A and B on McCoy cells (Table 3).

### 3.2. In Vivo Studies

In order to find an optimal CDI mouse model, BALB/c A and C57BL/6 female mice were tested. C57BL/6 mice were chosen for the study due to the outcome of CDI in C57BL/6 mice, which was more pronounced than BALB/c A mice. 

A pilot study with the mouse model of CDI confirmed the *C. difficile* infection in mice. It was shown that the fecal samples of the infected mice were infected by using both MALDI-TOF and PCR examination. A PCR specific for toxin tcdB confirmed replication of *C. difficile*.

The intestines of *C. difficile*-infected but untreated mice showed histological signs of ulcerative colitis with mucosal necrosis. These are typical histologic features of *C. difficile*-associated disease (Figure 2B). In contrast, *C. difficile*-infected and vancomycin-treated mice showed normal histology features (Figure 2A). Only one mouse in the *C. difficile*-infected and HXZQ-treated group showed slight-to-moderate segmental colitis.

The *C. difficile*-infected but untreated C57BL/6 mice exhibited weight loss (11% on Day 3) and a 50% death rate. Treatment with vancomycin protected mice against *C. difficile*-induced weight loss. One out of eight animals died during vancomycin treatment (presumably caused by a handling error). This single animal lost weight after treatment and died on day 9. Both groups of mice treated with HXZQ and a 1:1 diluted HXZQ lost weight in a similar fashion to the untreated controls. Seventy-five percent of the mice receiving HXZQ 0.1 mL twice a day survived, while 87.5% of the mice receiving a 1:1 saline dilution of HXZQ 0.1 mL twice a day survived. None of the surviving mice developed signs of relapse post-treatment (Figure 3 and Figure 4). There are no significant differences in body weight between the study groups analyzed by a two-sided *t*-test (Figure 4).

### 3.3. Combination of Vancomycin and HXZQ 

Figure 5 illustrates a weight loss of 9.62% in *C. difficile*-infected but untreated C57BL/6 mice on Day 5. In the vancomycin-treated group, the weight loss was 4.06% in mice on Day 8. As for vancomycin plus HXZQ 1:1 diluted with saline, in the treated mice, the weight loss was 3.08% on Day 8. No weight loss was observed in the HXZQ-treated animals. All groups of mice showed an increase in weight on Day 21 (one mouse, No. 37, died by accident on Day −1). That mouse (No. 37) was replaced by mouse No. 7 from the control one group. There are no significant differences in body weight between the study groups analyzed by a two-sided *t*-test (Figure 5).

Effects of vancomycin, HXZQ, and vancomycin plus HXZQ on the fecal *C. difficile* PCR loads of C57 BL/6 mice infected with *C. difficile* ATCC43255 are shown in the supplement result, Table 4. All the mice in the control 2 group and seven out of eight mice in the HXZQ alone group were *C. difficile* PCR-positive in their fecal samples two days after inoculation. Three out of eight mice in the vancomycin-treated group and four out of eight mice in the vancomycin plus HXZQ group were *C. difficile* PCR-positive in their fecal samples seven days after inoculation. All the vancomycin, HXZQ, and vancomycin plus HXZQ-treated mice were *C. difficile* PCR-positive in their fecal samples 13 days after inoculation. Four out of seven mice in the control 1 group were *C. difficile* PCR-positive in their fecal samples 21 days after inoculation (two of them were positive 13 days after). All the mice looked healthy and active on Day 21.
Effect of HXZQ on CDI in mice. Animals received HXZQ or vancomycin daily for five days from Day 4 and observed for 10 days from Day 0;Effect of combination HXZQ and vancomycin on CDI in mice. Animals received vancomycin or HXZQ daily for five days from Day 4 and were observed 21 days from Day 0.

## 4. Discussion

CDI is still the major cause of nosocomial infectious diarrhea, which affects approximately 500,000 patients annually in the United States. Of these, around 30,000 will die [19]. CDI remains a significant clinical challenge, both in the management of severe and severe-complicated disease and the prevention of recurrence [5]. Recently, fidaxomicin has been recommended by the IDSA/SHEA for the treatment of CDI [5]. However, in many countries, the main antimicrobial agent for the treatment of CDI is still vancomycin, and CDI recurrence is up to 35% of these cases. Other antimicrobial strategies, such as rifaxmin, are also recommended by the IDSA/SHEA. Furthermore, a number of studies showed that fecal microbiota transplantation and probiotics have good effects in the treatment of CDI [20,21,22,23]. Studies on other potential treatments of CDI, such as bacteriophage-derived endolysin, are ongoing [24].

Plant extracts have been used in Traditional Chinese Medicine to treat diarrhea since ancient China. HXZQ is one of the acclaimed and effective plant formulas for treating different kinds of diarrhea and other gastrointestinal diseases, such as, for example, functional dyspepsia.

The results of this study showed that vancomycin effectively protects mice against *C. difficile*-induced weight loss and mortality. HXZQ cannot protect mice against *C. difficile*-induced weight loss but did offer protection against *C. difficile*-caused mortality. Seventy-five percent of the mice receiving HXZQ 0.1 mL twice a day survived, while 87.5% of the mice receiving a 1:1 saline dilution of HXZQ 0.1 mL twice a day survived. One of the possible explanations for the better results with saline 1:1 diluted HXZQ is that HXZQ contains 45% alcohol. It has been shown that alcohol can damage the enteric mucosa [25]. When HXZQ is diluted with saline to 1:1, the concentration of alcohol is reduced to half and may not affect the enteric mucosa.

The results from the in vitro study showed that HXZQ does not kill *C. difficile* nor did it reduce the cytotoxicity induced by *C. difficile* Toxin A and B. The data from the fecal sample PCR indicate that HXZQ’s protection effect is not related to a direct antibacterial effect, since almost all the HXZQ-treated mice were *C. difficile* PCR-positive in their fecal samples three days after the infection. The mechanism of HXZQ-protective effects on mice against *C. difficile* infection is not clear, but it is tempting to speculate that this may be through the regulation of cytokines in the enteric mucosa, enhancing intestinal mucosal protection and protecting the intestinal mucosa from *C. difficile* infection. Histopathology shows less inflammation despite *C. difficile* infection in HXZQ-treated mice. Further investigation on the mechanism of action remains to be done.

The results from the in vivo study with a combination of vancomycin and HXZQ showed that the combination may give a better protective effect in *C. difficile*-infected mice. Even though all the mice were *C. difficile* PCR-positive in their fecal samples 21 days after the infection, they all looked healthy and active.

Recently, HXZQ has been shown to have anti-allergic effect properties [26], be able to prevent chemotherapy-induced nausea and vomiting [27], and has been used as a therapeutic agent, or in combination, for clinical colitis-associated cancer [28]. These results, together with Gao’s study [14] of HXZQ on gut-microbiome homeostasis indicate that HXZQ is a safe and well-tolerated agent. HXZQ can be used as a supplement or combination treatment for CDAD.

## 5. Conclusions

In conclusion, our results showed that HXZQ can protect mice from CDAD-related death as effectively as vancomycin, and a combination of vancomycin and HXZQ may give better protection in *C. difficile*-infected mice. This finding may provide a treatment option for patients who suffer from CDI.

## Figures and Tables

**Figure 1 microorganisms-12-01602-f001:**
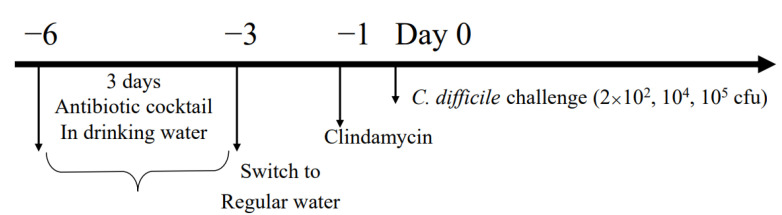
Experimental design.

**Figure 2 microorganisms-12-01602-f002:**
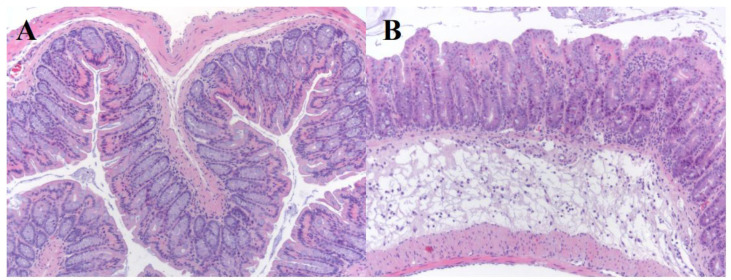
Histologic examination of colonic tissues on Day 3 from mice exposed to *C. difficile.* Mice treated with vancomycin show normal colonic mucosa (**A**). Untreated mice (saline solution) showed subacute enteritis with pronounced edema in submucosa with inflammatory cells (predominantly neutrophil) infiltrating both mucosa and submucosa (**B**).

**Figure 3 microorganisms-12-01602-f003:**
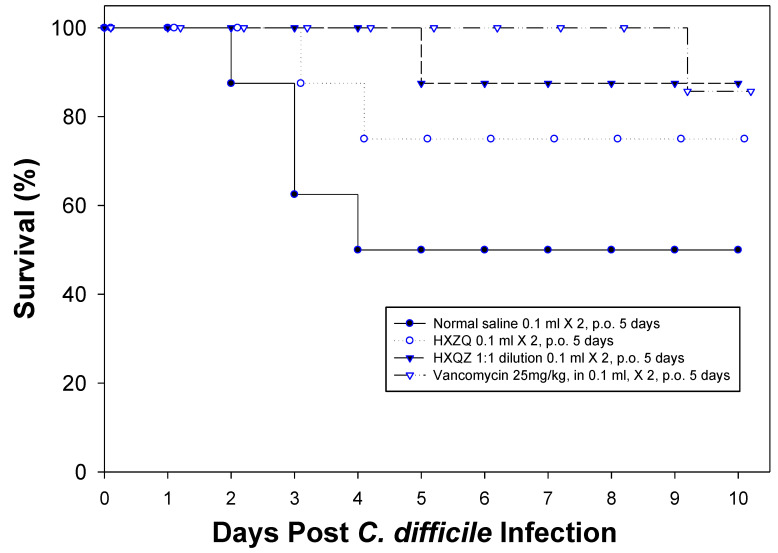
Survival rate of C57/BL6 mice after the infection of *C. difficile*. There were eight mice in each group. One of the vancomycin-treated mice died on day 3 because of a handling error.

**Figure 4 microorganisms-12-01602-f004:**
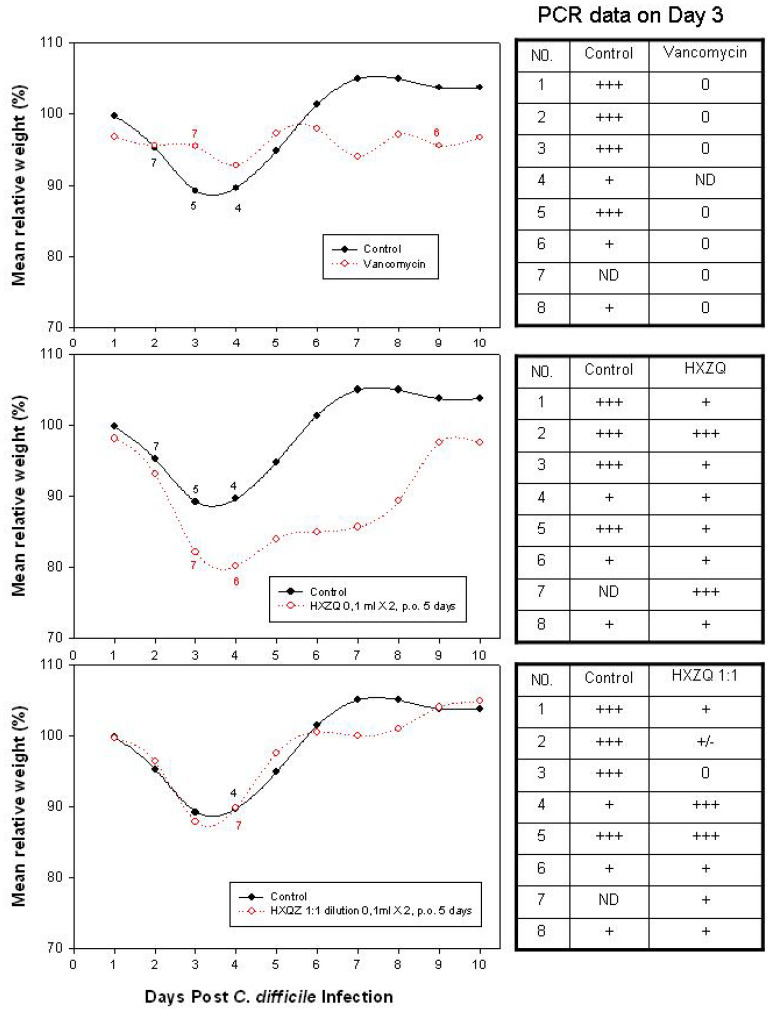
Relative body weight and fecal samples PCR data on Day 3. The numbers in the figure are number of mice in the groups on the day. PCR results: 0: negative; +: Ct > 30; +++: Ct < 30.

**Figure 5 microorganisms-12-01602-f005:**
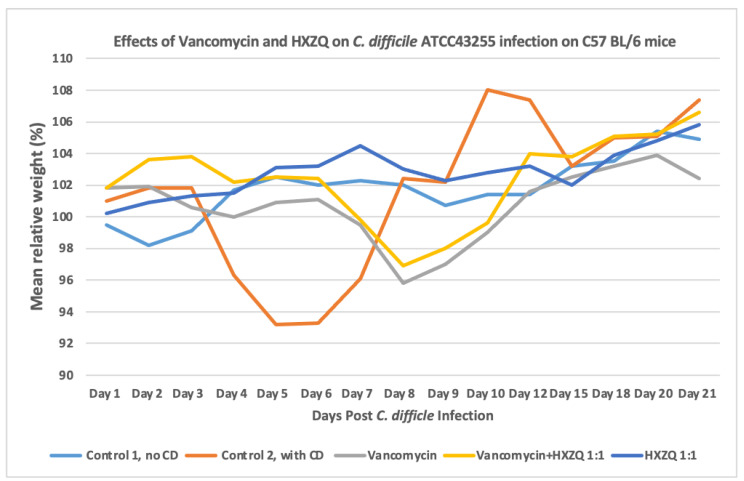
Effects of vancomycin, HXZQ, and vancomycin plus HXZQ on the relative body weights of C57 BL/6 mice infected with *C. difficile* ATCC43255. Eight mice were in each group.

**Table 1 microorganisms-12-01602-t001:** In vitro study of 20 μL of HXZQ or vancomycin or saline-diluted HXZQ on *C. difficile* ATCC43255 on a chocolate agar plate, a Mueller Hinton Agar (MHA) plate, and an SSI 5% blood agar plate. All agents were applied to paper discs (20 μL volume) on the tested plates. The inhibition zoon was measured in millimeters (mm). The results are given as inhibition zone (mm). The experiment was repeated three times.

	Chocolate Agar Plate	HMA Plate	SSI 5% Blood Agar Plate
1. vancomycin (20 μg)	26	20	26
2. HXZQ	8	8	10
3. HXZQ 1:1	6 *	6 *	0
4. Ethanol 45%	Not done	Not done	10

* Six mm corresponds to the diameter of the disc, which means zero inhibition.

**Table 2 microorganisms-12-01602-t002:** Effect of *C. difficile* (CD) bacterial supernatant and HXZQ on the in vitro growth of McCoy cells. The experiment was repeated three times.

	Optical Density	% of Control	% Inhibition
Control	958		
CD supernatant	660	69	31
HXZQ 1:64	730	76	24
HXZQ 1:128	860	90	10
HXZQ 1:256	920	93	7
CD Sup + HXZQ 1:64	670	70	30
CD Sup + HXZQ 1:128	720	75.2	25
CD Sup + HXZQ 1:256	700	73	27

**Table 3 microorganisms-12-01602-t003:** Effect of *C. difficile* Toxin A, B, and HXZQ on the in vitro growth of McCoy cells. The experiment was repeated three times.

	Optical Density	% of Control	% Inhibition
Control	1150		
Toxin A 200 ng/mL	620	54	46
Toxin A 200 ng/mL + HXZQ 1:128	680	59	41
HXZQ 1:128	980	85	15
Toxin B 200 ng/mL	700	61	39
Toxin B 200 ng/mL + HXZQ 1:128	750	65	35

**Table 4 microorganisms-12-01602-t004:** Effects of vancomycin, HXZQ, and vancomycin plus HXZQ on the fecal *C. difficile* PCR loads of C57 BL/6 mice infected with *C. difficile* ATCC43255. Eight mice were in each group.

Day 2	Control 1 (No *C. difficile*)	Control 2 (*C. difficile* ATCC43255)	*C. difficile* (ATCC43255) Vancomycin	*C. difficile* (ATCC43255) Vancomycin + HXZQ	*C. difficile* (ATCC43255) HXZQ
1	Negative	Positive +	Negative	Negative	Positive +++
2	Negative	Positive +	Negative	Negative	Positive +
3	Negative	Positive +++	Negative	Positive +	Positive +
4	Negative	Positive +	Negative	Negative	Positive +
5	Negative	Positive +++	Negative	Negative	Positive +
6	Negative	Positive +	Positive +	Negative	Positive +
7 *		Positive +	Negative	Negative	Positive +
8	Negative	Positive +	Negative	Negative	Positive +
**Day 7**	**Control 1 (No *C. difficile*)**	**Control 2 (*C. difficile* ATCC43255)**	** *C. difficile* ** **ATCC43255 Vancomycin**	** *C. difficile* ** **ATCC43255 Vancomycin + HXZQ**	** *C. difficile* ** **ATCC43255 HXZQ**
1	Negative	Positive +++	Negative	Positive +++	Positive +++
2	Negative	Positive +	Negative	Positive +++	Positive +
3	Negative	Positive +++	Negative	Positive +	Positive +
4	Negative	Positive +	Negative	Positive +++	Positive +
5	Negative	Positive +	Negative	Negative	Positive +++
6	Negative	Positive +++	Positive +	Negative	Positive +
7		Positive +++	Positive +	Negative	Positive +++
8	Negative	Positive +++	Positive +++	Negative	Positive +
**Day 13**	**Control 1 (No *C. difficile*)**	**Control 2 (*C. difficile* ATCC43255)**	** *C. difficile* ** **ATCC43255 Vancomycin**	** *C. difficile* ** **ATCC43255 Vancomycin + HXZQ**	** *C. difficile* ** **ATCC43255 HXZQ**
1	Positive +	Died	Positive +	Positive +	Positive +
2	Negative	Died	Positive +	Positive +	Positive +
3	Positive +	Positive +	Positive +	Positive +++	Positive +
4	Negative	Positive +	Positive +	Positive +	Positive +
5	Negative	Positive +	Positive +	Positive +++	Positive +
6	Negative	Died	Positive +	Positive +	Positive +
7		Died	Positive +	Positive +	Positive +
8	Negative	Died	Positive +	Positive +	Positive +
**Day 21**	**Control 1 (No *C. difficile*)**	**Control 2 (*C. difficile* ATCC43255)**	** *C. difficile* ** **ATCC43255 Vancomycin**	** *C. difficile* ** **ATCC43255 Vancomycin + HXZQ**	** *C. difficile* ** **ATCC43255 HXZQ**
1	Positive +	Died	Positive +	Positive +	Positive +
2	Positive +	Died	Positive +	Positive +	Positive +
3	Positive +	Positive +	Positive +	Positive +	Positive +
4	Positive +	Positive +	Positive +	Positive +	Positive +
5	Negative	Positive +	Positive +	Positive +	Positive +
6	Negative	Died	Positive +	Positive +	Positive +
7		Died	Positive +	Positive +	Positive +
8	Negative	Died	Positive +	Positive +	Positive +

* Mouse nr. 7 in *C. difficile* ATCC43255 HXZQ group died by accident before the inoculation of *C. difficile* on Day −1. Mouse nr. 7 in control 1 group was moved to *C. difficile* ATCC43255 HXZQ group as nr. 7. Therefore, there are only seven mice in control 1 group. PCR results: 0: negative; +: Ct > 30; +++: Ct < 30.

## Data Availability

Research data can be obtained by contacting the corresponding author of this paper.

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
