# Peer review of "Protective Effects of Huo Xiang Zheng Qi Liquid on Clostridioides difficile Infection on C57BL/6 Mice"

_microorganisms, 2024, doi:10.3390/microorganisms12081602_

Round 1

Reviewer 1 Report

Comments and Suggestions for Authors

iT IS AN ORIGINAL PAPER. tHE AUTHORS SHOWED THAT A TRADITIONAL CHINESE medicine (HXZQ) could have positive effects on Clostridium difficile infection on C57BL/6 MICE.Mice were firstly  challenged with C.difficile followed by saline (control), vancomycin and HXZQ. results showed the efficacy of HXZQ.

The paper is well designed and well written and all necessary ethical aspects were adressed.Bibliography is up to date .

Author Response

Response to Reviewer 1 Comments

1. Summary

2. Questions for General Evaluation

Reviewer’s Evaluation

Response and Revisions

Does the introduction provide sufficient background and include all relevant references?

Yes

Is the research design appropriate?

Yes

Are the methods adequately described?

Yes

Are the results clearly presented?

Yes

Are the conclusions supported by the results?

Yes

3. Point-by-point response to Comments and Suggestions for Authors

Response 1: Thank you very much for your valuable evaluation.

Reviewer 2 Report

Comments and Suggestions for Authors

Comments on the Quality of English Language

Author Response

The manuscript requires a series of modifications including spelling and editing of the text. In this format, the article should undergo major revision.

3. Point-by-point response to Comments and Suggestions for Authors

Comments 1:  Introduction section can be improved.

L 44 Please consider rephrase „ The excessive proliferation of C. difficile results in infection (CDI) characterized by symptoms such as diarrhea, fever, and pain. CDI is the predominant infectious cause of healthcare-acquired diarrhea globally and can lead to extended hospitalization [1, 3, 4].”

L46 Please consider rephrase „ This infection primarily affects elderly patients”

L 49 Please replace „ herbal medicines has been” with „ herbal medicines have been”

L 49 Please replace „ diarrhoea” with „ diarrhea” check for similar errors. Use the same term in the text.

L 50 Please consider rephrase „ A classical formulation, consisting of a mixture of various plant extracts known as "Huo Xiang Zheng Qi Liquid (HXZQ)," has been extensively used to treat a range of common gastrointestinal infectious diseases, including acute gastroenteritis and functional dyspepsia, since the Song Dynasty”

L55 „ Department of Song Dynasty in 1078-85” Unclear please rephrase (between 1078 and 1085?)

L 60 Replace „ enhancing” with „enhance”

Response 1: Thank you for pointing these out. We agree with the comment. Therefore, we have made the changes in the introduction section.

L44, L46, L 49, L50, L55 and L.60 have been rewritten according the reviewers suggestion. L 55 ”…Department of Song Dynasty in 1078-85 is changed to 1078” The first edition of the Prescriptions People’s Welfare Pharmacy (Taiping Huimin Hejiju Fang) was published 1078, but the final edition was published in 1151, even the formulation had been used many years before.

Moreover, L 71-75 have also been modified.

Comments 2: Methods must be improved

L77 Please remove excessive space between lines Yes. We did.

L 76 Please specify the producer, city and country for all the mentioned products/apparatus (infusion broth with reinforced 10% glycerol; PCR, MALDI-TOF). Please check for similar errors in the text. Yes. We correct these errors.

L 79 Please replace „ four days cultured C. difficile culture” with „ four days C. difficile culture” Yes. We replace it.

L 88 „ In vitro study of HXZQ on C. difficile ATCC43255 was first carried out in four different agar plates”, but only three agar types are mentioned: 1) Resistant plate, 2) Mueller Hinton II Agar (MHA) plate, and 3) SSI 5% horse 89 blood agar plate (5771422). Yes. We correct these errors.

L 89 What is „1) Resistant plate”? Please try to use appropriate description of the medium. Yes. We correct the error.

L 91 The sentence starts with a number „ 20 μl of vancomycin, ethanol or HXZQ were added on the top of discs.”; Please rephrase, the sentences should not start with numbers, check for similar errors in the text. (e.g. Twenty microliters of vancomycin). Yes. We correct these errors.

L 97 Please rephrase – similar to line 91 Yes. We correct the errors.

L 120-143 „ Mouse Model of CDI and Effect of HXZQ on CDI in mice” it’s hard to follow all of the information in the text, try to use different paragraphs for each idea. Yes. We have modified this section.

L 162 Please rephrase for more clarity Our pathologists have carefully checked this section.

Response 2: Agree. We have modified this section to emphasize this point as mentioned corrections above.

Comments 3:

Are the results clearly presented? Can be improved.

L 183-184 Please rephrase for more clarity

L 220 Table 1 – Unclear

Response 3: The authors agree on this. We have corrected the errors in L 183-184 and modified Table 1.

Comments 4: Statistical correlations among the groups are missing.

Response 4: Thank you for pointing them out.

Comments 5: Response to Comments on the Quality of English Language

Moderate editing of English language required

Response 5: Thank you for the point. We have done English language-proof-reading to correct errors.

Round 2

Reviewer 2 Report

Comments and Suggestions for Authors

Comments on the Quality of English Language

Author Response

The manuscript written by Ming Chen, Lin Zhai, Kristian Schønning, Warner Alpízar Alpízar, Ole Larum, Leif Percival Andersen, Susanne Holck and Alice Friis-Møller „Protective Effects of Huo Xiang Zheng Qi Liquid on Clostridioides difficile Infection on C57BL/6 Mice” is a good study, I still believe it requires a series of changes outlined below. The text also requires editing. Overall, there is an improvement of the manuscript.

Response: Thank you for pointing these out. We agree with the comment. Therefore, we have made the changes in L48, L72, L81, L92, L96, L100, L102 and L111 according the reviewers suggestion

L48 Please replace „The primarily affects” with „It primarily affects” Yes. We corrected it.

L72 „The aim of this study was to compare treatment of vancomysin with treatment of HXZQ and investigate the effect ofand diluted HXZQ 1:1 and combination of diluted HXZQ 1:1 plus vancomycin 7against CDI in a mouse model.” Please remove repeating word „and”, add space between words where needed; I also suggest to rephrase the sentence.

Yes, We rephrased the sentence, as “The aim of this study was to investigate HXZQ treatment alone and in combination with vancomycin treatment in a mouse CDI model and to compare their efficacy to standard vancomycin treatment.”

L81 Remove unnecessary space between line 81 and line 82. Yes, it has been removed.

L92 Please replace „ Chokolade agarplade” German description with „ Chocolate agar” English description, also replace in all text including tables. Yes. We have corrected it.

L96 Please replace „ 20 µl of vancomycin” with „ Twenty microliters of vancomycin” Yes, it has been changed.

L100 Use full name of the products „ RPMI1640 with 10% HFCS”, and add full description (producer, city, country) for each of them. Yes. We have corrected it.

L102 Please rephrase „ 2 X 105 cells in 0.2 ml were added into a well (96-wells plate) and incubated overnight”, all sentences must start with words not with numbers. Yes. It has been changed, as A cell culture was seeded at 2 X 105 cells in 0.2 ml in a well.”

L111 Ad description for „ cycloserine-cefoxitin fructose agar” (producer, city, country) Yes. We have corrected it.
